# The Proof-of-the-Concept of Application of Pelletization for Mitigation of Volatile Organic Compounds Emissions from Carbonized Refuse-Derived Fuel

**DOI:** 10.3390/ma12101692

**Published:** 2019-05-24

**Authors:** Andrzej Białowiec, Monika Micuda, Antoni Szumny, Jacek Łyczko, Jacek A. Koziel

**Affiliations:** 1Faculty of Life Sciences and Technology, Wrocław University of Environmental and Life Sciences, 50-375 Wrocław, Poland; micuda.monika@gmail.com; 2Department of Agricultural and Biosystems Engineering, Iowa State University, Ames, IA 50011, USA; koziel@iastate.edu; 3Faculty of Biotechnology and Food Science, Wrocław University of Environmental and Life Sciences, 50-375 Wrocław, Poland; Antoni.szumny@upwr.edu.pl (A.S.); jacek.lyczko@upwr.edu.pl (J.Ł.)

**Keywords:** RDF, CRDF, biowaste, waste to carbon, municipal solid waste, torrefaction, pyrolysis, VOCs, emissions, mitigation, pelletization, solid fuels, biochar, occupational safety

## Abstract

Waste can be effectively reused through the production of carbonized refuse-derived fuel (CRDF) that enables further energy recovery. Developing cleaner production of CRDF requires consideration of practical issues of storage and handling. Thus, it needs to be ensured that CRDF does not pose an excessive risk to humans and the ecosystem. Very few studies indicate a wide variety of volatile organic compounds (VOCs) are present in CRDF, some of which are toxic. During handling, storage, transportation, and use of VOC-rich CRDF, workers and end-users could be exposed to emissions that could pose a health and safety hazard. Our recent study shows that CRDF densification via pelletization can increase the efficiency of storage and transportation. Thus, the following research question was identified: can pelletization mitigate VOCs emissions from CRDF during storage? Preliminary research aiming at the determination of the influence of CRDF pelletization on VOCs emission during storage was completed to address this question. The VOCs emissions from two types of CRDF: ground (loose, torrefied refuse-derived fuel (RDF)) and pelletized, were measured. Pelletization reduced the VOCs emissions potential during the four-day storage by ~86%, in comparison with ground CRDF. Mitigation of VOCs emissions from densified CRDF is feasible, and research is warranted to understand the influence of structural modification on VOCs emission kinetics, and possibilities of scaling up this solution into the practice of cleaner storage and transportation of CRDF.

## 1. Introduction

The concept of ‘waste to carbon’ is the conversion of organic waste into valuable carbon materials, including fuels with a higher concentration of carbon [1]. For that purpose, the torrefaction process has been adapted to different types of organic waste [2,3,4,5]. Tests were carried out indicate that it is possible to convert municipal solid waste (MSW) into carbonized refuse-derived fuel (CRDF) with a calorific value between 24 and 26 MJ/kg, i.e., comparable to that of bitumen coal [6]. In another study [7], lower calorific value of refuse-derived fuel (RDF) increased due to torrefaction from 21.0 MJ·kg^−1^ to 25.9 MJ·kg^−1^. As well as this, Edo et al. [8] indicated the increase of RDF lower heating value from 19.7 MJ·kg^−1^ to 21.2 MJ·kg^−1^ due to 90 min torrefaction at 220 °C. CRDF addresses the problem of MSW management, in particular for MSW of organic origin. The waste can be transformed into CRDF, and thus it is not landfilled and could be used as fuel. 

The MSW torrefaction is one of the developing technologies, creating new, potential solutions for the ‘waste to energy’ concept [9]. In Poland, for example, the total annual production of MSW is ~12 mln Mg [10] of which ~2.6 mln Mg of RDF was produced in 2016 [11], while only ~1.2 mln Mg was thermally utilized with energy recovery [11] in cement kilns. The remainder of RDF (~1.4 mln Mg) is thermally treated in incineration plants or stored up for future processing. One of the undesired consequences of that is that the number of fires of unprocessed RDF almost doubled from 75 to 132 between 2012 and 2017 [10]. Such a situation enables research in waste (biomass)-derived carbon materials, including the development of novel technology of RDF torrefaction and production of CRDF with improved calorific value fuel properties. One of the requirements for torrefaction technology adoption is that it must be based on the sustainable approach, while the produced CRDF should be safe in handling and use.

One of the identified challenges related to the development of MSW torrefaction technology is the potential impact of biochar (including CRDF) on the environment, through emissions of volatile organic compounds (VOCs) [12,13] from carbon materials. Similar observations of VOCs emission occur during wood pellets storage, causing concerns of indoor air quality [14,15,16,17].

Moreover, the storage, transport, and use of combustible dust poses severe safety issues, as dust may explode when accidentally dispersed in air and ignited [18]. Thus, pelletization of CRDF is also useful to reduce the explosion risk [1]. Additionally, the emitted VOCs may be a source of self-ignition and pose a risk of fire during CRDF storage. However, Ceballos et al. [19] showed that torrefaction does cause depletion of volatiles components, which increases available surface active sites for oxidation (self-heating). Therefore, the self-heating propensity increases with fixed carbon content in biochar.

The development of the torrefaction—a new technology of waste treatment, shifting towards a recycling-based sustainable society, without evaluation of safety aspects, and embarking too readily on the practical application of newly developed CRDF, may trigger similar (fire) disasters, such as that which occurred at the refuse-derived fuel power station in Mie Prefecture in 2003 [20,21].

Similarly, the development of MSW torrefaction technology focused also on solving the problem of storage and transportation of CRDF, a bulky, powder-like material. Our recent work has shown that pelletization of CRDF can reduce its volume, thus improving the logistics of storage, transport, and use as fuel [1]. Pellets have the additional feature of being a standardized fuel that is easy to use in the fully automatic operation, simplifying the construction and operation of burners [22]. The benefits of biomass densification were widely presented and discussed in recent review papers [23,24,25], including the integration of torrefaction with pelletization [26]. Pelletizing technology is mature from production to end-use, and therefore, pelletized CRDF can help with adopting it for wide-scale use in waste management. Our recent research [27] shows that more than 80 VOCs are being emitted from CRDF, including toxic derivatives of benzene or toluene. Thus, practical solutions for the mitigation of VOCs emissions from CRDF are needed, especially applicable to storage and handling when workers are exposed to emitted compounds [28]. One of the solutions of VOC emission from CRDF may be pelletization.

Studies on VOC emissions during wood pelletization tend to focus on the work environment and measure concentrations of substances in the air at pelleting plants [29,30,31]. The quantitative estimations of emissions per ton of pellets produced have been explored only in a few studies [32,33]. Granström and Javeed [22] determined that during wood pellet production, only ~30–40% of the initial terpenes remain in the wood after dying, 20–30% remain after grinding, and 10%–15% remain after pelleting. It has been determined that wood pelletization itself causes the emission of VOCs [22]. However, the VOC emissions during CRDF pelletization have not been studied yet. 

In our recent study, we tested the effect of pre-treatment on VOCs emission from the CRDF in densified and bulk (powdery) form. We examined the influence of two scenarios of CRDF densification: pelletization, and pelletization with a binder (‘water glass’) on the kinetics of individual VOC emissions [34]. We determined that pelletization of CRDF was more effective in VOCs emission mitigation than grinding or pelletization with a binder. However, the important aspect is the application of CRDF pelletization to typical operational procedures of RDF torrefaction plants. As the RDF torrefaction technology is currently under development, a full-scale RDF torrefaction plant has not been in operation yet. However, it is reasonable to assume that within the near future, new technology will be implemented at full-scale. 

Therefore, proposed tests on VOCs emission rates from CRDF during storage may have great importance for preparing the investment plans of the future RDF torrefaction plants, including standard operating procedures, environmental impact assessments reports, and occupational safety and health protocols. It could open new research niches for in-depth understanding and development of fundamental knowledge about the relationship between the CRDF structure and its properties, and its implication in occupational safety and health protocols. As RDF torrefaction plants do not exist yet, by analogy, we applied to the typical conditions of RDF storage [35]. In the typical operation procedure at RDF production plants, the longest possible period of RDF storage is four days (e.g., RDF produced on Friday has to be in storage until Monday for transportation). Usually, the produced RDF, due to the risk of self-heating and, in consequence, fire and even explosion [20], is not stored longer than a couple of days. Similarly, wood pellets storage is not recommended to be longer than several days [36]. By analogy, in this paper, we focused on the proposed four days of CRDF storage as being the ‘worst case’ scenario for a production facility and explored VOC emissions. 

For this reason, we proposed to compare the VOCs emission rates from the ground and pelletized CRDF during four days of storage and to identify the VOCs associated with the highest emissions and VOCs posing imminent danger to safety and health [37]. As the next step of RDF torrefaction technology, the development from fundamental to applied research of the simulation of VOCs concentration in the headspace of the storage bin was proposed, with a comparison of the Polish and USA occupational health standard threshold values for inhalation of VOCs. This may bring new, important technological requirements for CRDF storage.

## 2. Materials and Methods 

### 2.1. CRDF Samples

The RDF used for CRDF production was collected from a mechanical-biological waste treatment facility with the status of a regional waste treatment plant. The facility is located in the village of Gac, Poland (in the region of Lower Silesia). The detailed process description of RDF production and RDF properties were presented by Stepien and Bialowiec [38]. The subject of the research was CRDF produced from RDF due to the torrefaction under 260 °C, and with the retention time 50 min in a batch reactor, according to the procedure used by Białowiec et al. [1]. The resulting lower heating value of CRDF was ~27.3 MJ·kg^−1^. The description of the properties of produced CRDF was shown in detail by Białowiec et al. [1].

Briefly, two variants of CRDF were used for the VOCs emission test:ground (loose powder);pelletized.

The produced CRDF was ground in the LMN-100 crushing mill (Testchem, Pszów, Poland) on a 1 mm diameter sieve. In this bulky form, CRDF was used in this experiment as the first variant.

We subjected ground CRDF to pelletization tests using a controlled pressure of 50.8 MPa. The pelletization of CRDF was carried out by compacting the material using the INSTRON 5566 (Instron, Norwood, MA, USA) testing machine at ambient temperature [1]. In this pelletized form, CRDF was used in this experiment as the second variant.

### 2.2. Qualitative and Quantitative Analyses of VOC Emitted from CRDF

The whole procedure of determination of VOCs emission from 10 g samples of CRDF—ground and pelletized—was completed using the identical conditions as presented in Białowiec et al. [11]. VOCs emissions from two types of processed CRDF were measured after four days of storage in 1 L glass vessels. The glass vessels with material were kept in constant temperature in an incubator at 23 °C. The four day emission duration was chosen to simulate the ‘worst case’ scenario of storage over a long weekend at a small batch CRDF production facility.

The emitted VOCs were extracted from the vessels’ headspace by SPME (solid-phase microextraction) method [27]. Briefly, gas samples were collected with the three-component SPME fiber coating (DVB/CAR/PDMS 50/30 μm) that was introduced into the headspace of the sealed vessel with the biochar sample. During SPME extraction, the vessel was held in a water bath with glycol preheated to 40 °C and kept in this temperature for the whole extraction time, which was 20 min. The used DVB/CAR/PDMS 50/30 μm SPME coating is often recommended and used for exploratory work on VOC emissions from unknown sources [27,39,40,41].

The gas chromatography with mass spectrometry (GC-MS) was used for separation, identification, and quantification (2-undecanone was used as the internal standard) of VOCs with the procedure described in Białowiec et al. [11]. Briefly, the separation, identification, and quantification of VOCs adsorbed on the fiber was conducted using a gas chromatography (GC) coupled to a mass spectrometry (MS) detector (Saturn 2000 MS Varian Chrompack, Palo Alto, CA, USA) with a ZB-5 (Phenomenex, Torrance, CA, USA) column (30 m × 0.25 µm film × 0.25 mm inside diameter (i.d.)). Chromatographic conditions were performed according to Calin-Sanchez et al. [42]. Scanning (1 scan/s) was performed in the range of 35–400 m/z (mass-to-charge ratio) using electron impact ionization at 70 eV [43]. The analyses were performed using helium as a carrier gas at a flow rate of 1.0 mL/min, in splitless mode in SPME, and with the following program for the oven temperature: 50 °C at the beginning; 4 °C/min to 130 °C; 10 °C/min to 180 °C; and 20 °C/min to 280 °C with a hold for 4 min. The injector was held at 220 °C. The blank was done on the same day to evaluate the current experiment conditions and the possibility of interferences as room air was used during the experiment. Additional information was added to the manuscript as an attachment (i.e., a blank sample (SMS format (mass spectral data format (Bruker/Varian instrument data format))).

### 2.3. Evaluation of VOC Emissions Mitigation Potential

The use of an internal standard enabled quantitative analysis of VOCs, which were then compared. The mitigation rate (%) for each of the 84 VOCs emitted from stored, pelletized CRDF was estimated in relation to ground CRDF (control). Figure 1 presents the applied experimental procedure.

### 2.4. The Simulation of VOC Concentration in the Air during the Four Day Storage of Ground or Pelletized CRDF

A new model for the simulation of VOCs concentrations was prepared in an Excel spreadsheet (Appendix A). The following assumptions were made:The volume of CRDF fills 50% of the total volume of storage bin;The volume of the storage bin is 1 m^3^;The bulk density of ground CRDF is 424.4 kg·m^−3^ [1];The bulk density of pelletized CRDF is 625.0 kg·m^−3^ [44]—it has been assumed that CRDF pellets’ bulk density is equal to the average value from the most commonly found pellets’ bulk density range between 550 and 700 kg·m^−3^.

The calculated concentrations of particular VOCs (only VOCs having health standard threshold values were chosen for calculations) were compared with Polish and USA health standards of VOCs’ threshold values and inhalation exposure. The following standards were used:HAC—The highest allowed concentration [45]—Polish standard;HAMC—The highest allowed momentary concentration [45]—Polish standard;IDLH—Immediately dangerous to life or health [46]—USA standard;STEL—Short term exposure limit [46]—USA standard;TWA NOISH—Time weighted average concentration based on National Institute for Occupational Safety and Health (NIOSH) REL for a 10 h workday —NIOSH REL (recommended exposure level) (this is ‘recommended’) time-weighted average (TWA), ‘up to 10 h exposure limit during a 40 h workweek’ [46]—USA standard,TWA OSHA—Time weighted average concentration or Occupational Safety and Health Administration (OSHA) PEL for 8 h TWA concentration) OSHA PEL (permissible exposure limits) (this is regulatory) TWA, 8 h exposure limit [46]—USA standard.

The fold of exceedance of the standard threshold values was calculated by dividing the estimated VOC concentrations by standard threshold values. The model and data used for simulations are available in the Appendix A as ‘Model of VOC emissions from CRDF during storage.xlsx.’.

## 3. Results

Comparisons of VOC emissions potential for 84 compounds (mass of VOC emitted per mass of CRDF) and the % mitigation rate are summarized in Table 1. 

Identified compounds have been organized to five groups: (A)alkyl derivatives of benzene or phenols (possible carcinogens) (32 compounds with an emission of 4196.92 μg/kg (ground) and 576.28 μg/kg (pelletized));(B)alkyl derivatives of two-ring aromatic hydrocarbon (16 compounds with an emmision of 1367 μg/kg (ground) and 178 μg/kg (pelletized));(C)derivatives of heterocyclic amines (7 compounds, with an emission of 170 μg/kg (ground) and 23 μg/kg (pelletized));(D)compounds that are generally considered as lower risk (e.g., present naturally in food) (22 compounds with an emission of 8097 μg/kg (ground) and 1113 μg/kg (pelletized));(E)belonging to other groups or with an unknown structure (7 compounds with an emission of 662 μg/kg (ground) and 95 μg/kg (pelletized).

According to [37], the 11 compounds were identified as immediately dangerous to life or health (IDLH), these being emissions 1196 (8.25% of total emission) and 157 (7.86% of total emission) μg/kg from ground and pelletized forms, respectively. Moreover, 52 compounds have a flash point below 100 °C (emissions 142 (0.98% of total emission) and 20 (1.0% of total emission) μg/kg from ground and pelletized form, respectively), while 3 of them have flash points in the range of 100–150 °C. 

In the case of 37 of the compounds: the lower explosive limit (LEL) was identified [37], and was present in emission rates 10420 (71.84% of total emission) and 1424 (71.31% of total emission) μg/kg before and after pelletization, respectively.

The emissions of the mentioned compounds causes the risk of a toxic effect on humans, but also the risk of self-ignition, and in consequence fire or explosion. Therefore, the problem of the control and mitigation of the emissions of these compounds from CRDF should be solved. One of the proposed methods is pelletization. 

Pelletization reduced the total VOC emission rate by 86% for four days of CRDF storage. In most cases (except for five VOCs) the pelletization reduced the emission rate. This preliminary study revealed that VOCs emission mitigation due to pelletization was highly effective. Emissions of 75 of the identified VOCs were reduced with an efficiency higher than 80%. One of the potential explanations could be the decrease of the contact surface area by densification of the CRDF structure. Pelletization likely decreases the porosity and VOCs adsorbed, or those present inside pores are displaced from CRDF as pore space becomes smaller. Our findings are consistent with the observation of the influence of wood pelletization on VOCs emissions. It has been shown that during wood pellet production, only ~30–40% of the initial terpenes remain in the wood after dying, 20–30% remain after grinding, and 10–15% remain after pelletization [22]. It is also known that wood undergoes several chemical changes during the process of pelletization. Roffael and Kraft [47] determined the change in the pH value of wood during the different steps of pellet making and established that the pH value of spruce wood decreases noticeably, and the alkaline buffering capacity increases significantly during the process of pellet making. Moreover, the amount of cold-water extractives of spruce wood increases, indicating some hydrothermal degradation of wood during the process of pellet making. Especially, the pressing process at high temperature and pressure appears to thermally degrade wood [36]. 

In the present study, CRDF produced due to torrefaction from RDF (a mixture of a flammable fraction of MSW: plastics, paper, cardboard, different textiles, and biowaste) was subjected to pelletization but under ambient temperature. Some heat could be released due to friction phenomena during pelletization, which could cause thermal degradation of VOCs present in bulky material or simply a release of VOCs due to volatilization under increased temperature. The nature of these phenomena is unknown, however, this aspect is very interesting and may be a subject of the next research project.

Additionally, it has been found that five VOCs increased the emission rate due to pelletization, including styrene (a possible carcinogen). The observed process of reduction and generation of VOC could be related to the changes in physical properties, such as the permeability of the matrix. Volatile compounds could evaporate from pelletized biomass, with a more solid surface in a relatively different efficacy, which was found in case of pentanoic acids, for example. Similar observations (i.e., a higher emission of selected compounds from the dense matrix) were found in the case of preparation of herbal comprimates (products of the pressure agglomeration process) [48] or during the drying process [43], which affects the aroma profile of food [49]. The mechanism and factors influencing VOCs emission that increase as caused by pelletization of different matrices are not recognized and should be studied.

The presented results are novel and open a new niche for investigations and experiments. Particularly from the fundamental science point of view, the mechanism of VOCs generation during torrefaction, the influence of the type of waste feedstock for torrefaction on the presence of VOCs in biochar, and the mechanism of VOCs mitigation during palletization, are intriguing and should be further investigated.

Several of the compounds identified in emissions from CRDF are considered as low risk and non-toxic and often occur as volatile food ingredients. For example, 2- and 3-methylbutanoic, propanoic acid as well as heptanone occur in cheese [51], 2-methoxyphenol and pentan-1-ol occur in wine [52], non-cyclic saturated aldehydes (hexanal, octanal, nonanal, decanal) occur in brown millet [53], and benzaldehyde occurs in cherry [54]. On the other hand, some furan compounds are well-known as products of the Maillard thermal degradation of food (i.e., furan-2-carbaldehyde and 5-methylfuran-2-carbaldehyde [55]) or hydrothermal degradation of compounds from biomasses rich in pectin [56]. Methyl benzoate could be found as a volatile compound in Styrax benzoin. 

The conducted simulation of VOCs concentration in the headspace of the storage bin showed that in the scenario with ground CRDF, the health standard threshold values were exceeded in the case of acetic acid (USA standards), toluene (Polish standards), furan-2-carbaldehyde (USA standards), 1,3,5-trimethylbenzene (Polish and USA standards), and phenol (Polish and USA standards) (Appendix A ‘Model of VOC emissions from CRDF during storage.xlsx—spreadshhet ‘Ground CRDF’’). The fold of standards’ exceedance was the highest in the case of furan-2-carbaldehyde, which was higher than 20. This finding might be considered as a practical implication for designing the forced ventilation of the ground CRDF storage bin and/or the production hall. 

In the case of pelletized CRDF, the VOC emissions simulation revealed that pelletization significantly reduces the VOCs concentrations in the storage bin. It has been found that only furan-2-carbaldehyde has the potential to exceed the standard threshold value (Appendix A ‘Model of VOC emissions from CRDF during storage.xlsx’—spreadsheet ‘Pelletized CRDF’). The fold of exceedance was above four, and could be considered for improved design of forced ventilation of storage bins. Most likely, much lower ventilation rates would be needed for pelletized CRDF, which could represent cost savings. Another solution could be decreasing the volume of CRDF to the headspace volume ratio in the storage bin, a solution that likely could increase the overall costs. The provided Excel spreadsheet allows such simulation, and may be used for further experiments in that field.

These findings have great importance related to practical considerations for the mitigation of VOCs emissions via pelletization. Pelletization improves storage conditions (decrease of volume) and more importantly, the reduction of occupational exposure to VOC emissions from CRDF. Our experiment showed the global effect of the influence of CRDF pelletization on VOCs emission. The release of VOCs during pelletization could be the reason why the emissions measured were lower, but this needs to be investigated in the future. The pelletization reduces also the risk of health standards threshold values exceedance and influences on the technical requirements for forced ventilation of the storage bin.

Our experiment was rather exploratory in this field and aimed at testing the practical approaches to VOC mitigation by pelletization. To avoid harmful human exposure and risk of fire and explosion (including dust explosion), it is important to monitor and, if necessary, take actions to reduce VOC off-gassing from different CRDF production steps where exposure might occur [57]. It should be noted that except in the case of VOCs emission, the formation of H_2_ and CH_4_ formation during CRDF storage should be monitored. Potential VOCs emissions during pelletization can be controlled by typical industrial approaches, such as vents and hoods that are applicable to localized workplace sources of air pollutants. The provided modeling spreadsheet could be improved by, for example, a more advanced model for predicting the VOCs concentration in the air during storage for practical, full production scale and production facility implementation. Further studies should be continued in this field. Emissions during pelletization should also be investigated on an industrial scale. 

We recommend continuing studies on the mechanism of VOC emissions, including the kinetics of VOCs emission from CRDF. Also, the effects of other types of CRDF biochars on VOC emissions should be studied. In our opinion, this opens a wide niche for investigation both in fundamental science, such as an explanation of the VOCs emission mechanism, and applied science, such as the scaling up of the system for investigating the potential impact of VOCs on workers during CRDF storage and methods of mitigation of VOCs emission as a part of MSW torrefaction technology development. The biochar technology production and its application in the environment is under significant development. However, there is a “dark side” of the biochar related to its potential toxicity and emission of pollutants into the environment. Studies in that field should not only focus on biochars from RDF, but also from other organic waste, such as sawdust, sewage sludge, dairy manure, poultry manure, and dried distillers grains with solubles in relation to different torrefaction and pyrolysis temperatures.

## 4. Conclusions

CRDF production from RDF is a novel method of MSW fuel valorization and production of the new type of carbonized fuel. The presented preliminary research on the influence of pelletization on VOCs emission from CRDF show that:VOCs are emitted from CRDF during storage.Pelletization reduced the VOCs emissions potential during storage by ~86%, in comparison with ground CRDF.Due to pelletization, it was possible to reduce the emission of 10 highly toxic compounds, 50 highly flammable compounds, and 35 potentially explosive VOCs.CRDF pellets can be classified as a less hazardous material than bulky and powdery CRDF.CRDF pelletization reduces the risk of exceeding health standard threshold values for occupational inhalation exposure and influences on technical requirements of forced ventilation of CRDF storage bins or a production facility.

Mitigation of VOC emissions from densified CRDF is confirmed, and research is warranted to understand the influence of structural modification on VOCs emission kinetics, as well as possibilities of implementing of this solution into the practice of cleaner production of pellets, volume reduction, storage, transportation, and utilization logistics of CRDF. Further research on the mechanism and kinetics of VOCs emission from CRDF and other biochar types is recommended. Emissions during pelletization should also be investigated on an industrial scale.

## Figures and Tables

**Figure 1 materials-12-01692-f001:**
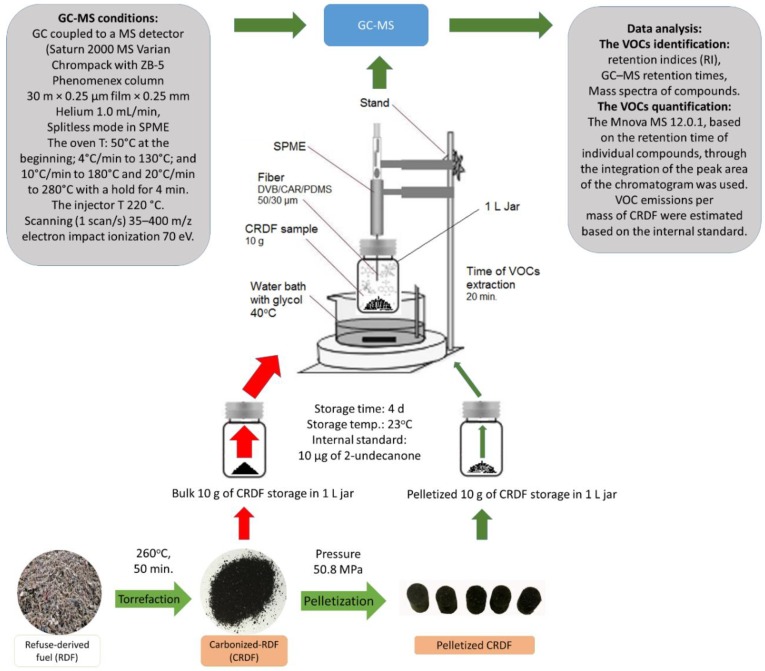
The experimental procedure used to measure VOC emissions from stored carbonized refuse-derived fuel. GC-MS—gas chromatography with mass spectrometry, GC—gas chromatography, MS—mass spectrometry, SPME—solid-phase microextraction, VOCs—volatile organic compounds, CRDF—carbonized refuse-derived fuel.

**Table 1 materials-12-01692-t001:** VOCs emitted from torrefied carbonized refuse-derived fuel (CRDF): ground and pelletized.

Compound Name, International Union of Pure and Applied Chemistry (IUPAC)	Chemical Abstracts Service (CAS) Number	Emission (µg/kg)		Emission Mitigation rate (%)	IDLH [37]	Flash Point [37]	Lower Explosive Limit [37]
		Ground CRDF	Pelletized CRDF	Pelletized CRDF	mg·dm^−3^	°C	%
3-methylbutanoic acid (D)	503-74-2	1.55	0	100	nd	21.1	1.5
4-ethylpyridine (C)	536-75-4	4.65	0	100	nd	50.0	nd
5-methyl-1,2,3,4-tetrahydronaphthalene (B)	2809-64-5	0.01	0	100	nd	nd	nd
1,2-xylene (A)	95-47-6	43.41	0.9	97.9	3906	32.2	0.9
heptan-2-one (D)	110-43-0	116.28	3.3	97.2	3736	32.9	1.1
1,3-diethylbenzene (A)	141-93-5	119.38	4.79	96.0	nd	51.0	0.8
2,5-dimethylpyrazine (C)	123-32-0	6.2	0.3	95.2	nd	64.0	nd
5-methyl-2,3-dihydro-1H-indene (B)	874-35-1	89.92	5.39	94.0	nd	nd	nd
4-methyl-2,3-dihydro-1H-indene (B)	824-22-6	122.48	7.49	93.9	nd	nd	nd
1,4-dimetylopirydyne (C)	108-47-4	23.26	2.1	91.0	nd	50.0	nd
Hexylbenzene (A)	1077-16-3	451.16	42.55	90.6	nd	83.0	0.7
pyridine (C)	110-86-1	6.2	0.6	90.3	3240	20.0	1.8
1-(furan-2-yl)ethenone (E)	1192-62-7	43.41	4.79	89.0	nd	71.0	nd
2-ethylpyrazine (C)	13925-00-3	12.4	1.5	87.9	nd	23.0	nd
2,4-diethyl-1-methylbenzene (A)	1758-85-6	12.4	1.5	87.9	nd	nd	nd
2-ethyl-1,4-dimethylbenzene (A)	1758-88-9	55.81	6.89	87.7	nd	58.9	nd
acetic acid (D)	64-19-7	148.84	18.58	87.5	123	39.4	4.0
2,3-dihydro-1H-indene (B)	496-11-7	54.26	6.89	87.3	nd	50.0	nd
heptanal (D)	111-71-7	1051.16	134.55	87.2	nd	35.0	1.1
decanal (D)	112-31-2	458.91	58.74	87.2	nd	83.0	0.8
propanoic acid (D)	79-09-4	13.95	1.8	87.1	nd	52.2	2.9
*n*-propylbenzene (A)	103-65-1	27.91	3.6	87.1	nd	42.0	0.9
1,2-diethylbenzene (A)	135-01-3	88.37	11.39	87.1	nd	55.0	0.8
octanal (D)	124-13-0	1244.96	162.12	87.0	nd	52.0	1.0
an unknown isomer of ethyl-dimethyl benzene (A)	-	85.27	11.09	87.0	nd	nd	nd
4-ethyl-1,2-dimethylbenzene (A)	934-80-5	85.27	11.09	87.0	nd	nd	nd
1-methyl-1H-indene (B)	767-59-9	200	26.07	87.0	nd	77.0	nd
1,3,5-trimethylbenzene (A)	108-67-8	327.13	42.85	86.9	nd	50.0	0.9
1,3-xylene (A)	108-38-3	93.02	12.29	86.8	3906	27.8	1.1
2-ethyl-1,4-dimethylbenzene (A)	1758-88-9	113.18	14.98	86.8	nd	58.9	nd
Phenol (A)	108-95-2	65.12	8.69	86.7	962	79.4	1.8
2-methyl-5-propan-2-ylphenol (B)	499-75-2	114.73	15.28	86.7	nd	nd	nd
2-methylpyrazine (C)	109-08-0	31.01	4.2	86.5	nd	50.0	nd
unknown compound (E)	-	51.16	6.89	86.5	nd	nd	nd
1,2,3,5-tetramethylbenzene (A)	527-53-7	97.67	13.19	86.5	nd	63.0	0.8
6-methyl-1,2,3,4-tetrahydronaphthalene (B)	1680-51-9	57.36	7.79	86.4	nd	259.7	nd
dec-3-yn-1-ol (E)	51721-39-2	63.57	8.69	86.3	nd	107.0	nd
2-ethyl-1,3-dimethylbenzene (A)	2870-04-4	220.16	30.27	86.3	nd	nd	nd
nonanal (D)	124-19-6	2443.41	338.93	86.1	nd	64.0	0.8
Pentylbenzene (A)	538-68-1	66.67	9.29	86.1	nd	66.0	0.7
furan-2-carbaldehyde (D)	98-01-1	96.12	13.49	86.0	393	60.0	2.1
1-ethyl-3,5-dimethylbenzene (A)	934-74-7	212.4	29.67	86.0	nd	57.3	nd
1-undecyne (E)	2243-98-3	330.23	46.15	86.0	nd	65.0	0.7
Undecane (E)	1120-21-4	117.83	16.48	86.0	nd	60.0	0.7
2-oxopropyl acetate (E)	592-20-1	68.22	9.59	85.9	nd	nd	nd
1-methyl-4-prop-1-en-2-ylcyclohexene (D)	138-86-3	737.98	105.78	85.7	nd	nd	nd
1,2,3,4-tetrahydronaphthalene (B)	119-64-2	91.47	13.19	85.6	nd	71.0	0.8
4,6,6-trimethylbicyclo[3.1.1]hept-3-ene (D)	80-56-8	365.89	53.04	85.5	nd	nd	nd
5,6-dimethyl-1,2,3,4-tetrahydronaphthalene (B)	21693-54-9	12.4	1.8	85.5	nd	nd	nd
2-ethyl-2,3-dihydro-1H-indene (B)	56147-63-8	272.87	40.16	85.3	nd	nd	nd
1,4-xylene (A)	106-42-3	178.29	26.37	85.2	3960	27.2	1.1
4,7-dimethyl-2,3-dihydro-1H-indene (B)	6682-71-9	212.4	31.47	85.2	nd	nd	nd
1,5-dimethyl-1,2,3,4-tetrahydronaphthalene (B)	21564-91-0	26.36	3.9	85.2	nd	106.1	nd
1-ethenyl-2,4-dimethylbenzene (A)	2234-20-0	100.78	15.28	84.8	nd	nd	nd
Butylbenzene (A)	104-51-8	903.88	138.45	84.7	nd	59.0	0.8
1,3-dimethyl-2,3-dihydro-1H-indene (B)	4175-53-5	103.88	15.88	84.7	nd	nd	nd
toluene (A)	108-88-3	448.06	68.92	84.6	1885	4.4	1.1
hexanal (D)	66-25-1	652.71	100.69	84.6	nd	25.0	1.3
1-phenylethanone (A)	98-86-2	234.11	36.26	84.5	nd	77.0	nd
azulene (B)	275-51-4	15.5	2.4	84.5	nd	nd	0.9
1,3-diethyl-5-methylbenzene (A)	2050-24-0	32.56	5.09	84.4	nd	nd	nd
Benzaldehyde (D)	100-52-7	548.84	87.8	84.0	nd	64.0	1.4
4-methyl-1-propan-2-ylcyclohexene (E)	500-00-5	88.37	14.38	83.7	nd	nd	nd
methyl benzoate (D)	93-58-3	49.61	8.09	83.7	nd	83.0	1.2
pentan-1-ol (D)	71-41-0	20.16	3.3	83.6	nd	49.0	1.2
1,2,4-trimethylbenzene (A)	95-63-6	10.85	1.8	83.4	nd	44.4	0.9
pyrimidine (C)	289-95-2	86.82	14.68	83.1	nd	34.0	nd
an unknown isomer of diethyl methylbenzene (A)	-	17.05	3	82.4	nd	nd	nd
1,2,4,5-tetramethylbenzene (A)	95-93-2	82.17	14.98	81.8	nd	74.0	0.8
1-methyl-4-propan-2-yl-2-[(*E*)-prop-1-enyl]benzene (A)	97664-18-1	20.16	3.9	80.7	nd	nd	nd
cumene (A)	98-82-8	1.55	0.3	80.6	4428	35.6	0.9
5-methylfuran-2-carbaldehyde (D)	620-02-0	6.2	1.2	80.6	nd	72.0	nd
2-methoxyphenol	90-05-1	6.2	1.2	80.6	nd	82.0	1.3
unknown compound (E)	-	3.1	0.6	80.6	nd	nd	nd
1-methylnaphtalene (B)	90-12-0	1.55	0.3	80.6	nd	82.0	nd
3,3-dimethyl-2H-inden-1-one (B)	26465-81-6	1.55	0.3	80.6	nd	nd	nd
unknown compound (E)	-	27.91	7.19	74.2	nd	nd	nd
1-methyl-2-propylbenzene (A)	1074-17-5	3.1	0.9	71.0	nd	nd	nd
2-methylpropanoic acid (D)	79-31-2	1.55	0.6	61.3	nd	55.0	nd
undecan-2-one (internal standard) (E)	112-12-9	1000.00	1000.00	0.0	nd	nd	nd
pentanoic acid (D)	109-52-4	1.55	9.29	−499.4	nd	96.0	1.6
1,4-diethyl-2-methylbenzene (A)	13632-94-5	0.01	0.3	−2900.0	nd	nd	nd
hexa-2,4-diene, (*E,E*)- (D)	592-46-1	0.01	1.5	−14,900.0	nd	nd	nd
1-methyl-4-propan-2-ylbenzene (A)	99-87-6	0.01	2.1	−20,900.0	nd	47.0	nd
Styrene (A)	100-42-5	0.01	3.6	−35,900.0	2982	31.1	0.9
Total	-	14,503.88	1996.70	86.2			

The identified groups of compounds were classified to five groups marked by colors: A—alkyl derivatives of benzene or phenols carcinogens [50]; B—alkyl derivatives of two-ring aromatic hydrocarbons; C—derivatives of heterocyclic amines; D—compounds that are generally considered as lower risk (e.g., present naturally in food); E—other or unknown. nd—no data. ILDH—immediately dangerous to life or health.

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
