# Peer review of "The Proof-of-the-Concept of Application of Pelletization for Mitigation of Volatile Organic Compounds Emissions from Carbonized Refuse-Derived Fuel"

_materials, 2019, doi:10.3390/ma12101692_

Reviewer 1 Report

The manuscript titled "The mitigation of VOC emmissions form CRDF storage with pelletization" by A. Bialowiec et al. presents a comparison of the emissions of two techniques of CRDF storage : ground and pelletized CRDF. This work is a continuation of a previous work published recently. The manuscript lists the VOC observed from CRDF after a four days storage as identified using a SPME technique. An expected decrease of emissions during storage is observed between ground and pelletized CRDF.
Unfortunalty, I don't consider this work to present sufficient novelty and originality to be accepted as is. As, I think, an important amount of work would be necessary to improve the quality of the manuscript, I recommend rejection in the present form.

My main concerns are :
- the novelty of the results : the technique and emissions of a CRDF as already been published recently. Next publication on the same subject and using the same technique should then add an important value as all technical aspects (analytical in particular) have already been set. The work presents only two experiments. For instance, replications should be made, and tests of different kind of CRDF, other storage techniques... Furthermore, the decrease in VOC emission from pelletization is an expected result and does not present sufficient interest to be published in a communication format.
- Method description : that part could be improved by presenting rapidly what was the main points of the previous publications instead of refering to them. It would be a help to the reader.
- Results discussion : the discussion does not appear to be developped enough. The only discussion concerns the total VOC mitigation rate, there is nothing on the type of VOC emitted nor explanation on their presence/absence, and more interestingly on the formation of new VOC after pelletization (where do they com from ?). Such as discussion should be added.
Moreover, as the goal is to reduce the overall toxicity of the emission, there could be a discussion on the toxicity of the newly observed compounds.  
A comparison to the VOC emitted during pelletization process may add an interesting value to the work to have a global knowledge of the global process.
- Introduction : The introduction does not put into more general perspective the interests of the study for the reader of the journal. As their is a large use of self-citation to present the interest of the work, I have a concern on the significance of the work for the community. It then should be rewritten to broaden the audience.

Author Response

We included detail responses to the reviewer in the attached file.

Reviewer 2 Report

This is a nice paper addressing an interesting topic. It is also well written and organized. As such, it can deserve publication in Materials. However, prior to publication, I have some comments that the authors should scrupulously implement into the revised manuscript.

1) Introduction - The authors should include the following sentences

“Moreover, the storage, transport, and use of combustible dusts pose severe safety issues, as such dusts may explode when accidentally dispersed in air and ignited [xx]. Thus, pelletization of CRDF is also useful to reduce the explosion risk.” - with [xx] = Chemical Engineering Transactions, Volume 31, 2013, Pages 727-732

between

“Similarly, the development of MSW torrefaction technology focused also on solving the problem of storage and transportation of CRDF, a bulky, powder-like material. Our recent work has shown that pelletization of CRDF can improve the fuel properties of the product and reduce its volume, thus improving the logistics of storage, transport, and use as fuel [1].”

and

“The benefits of biomass densification were widely presented and discussed in recent review papers [13-15] including integration of torrefaction with pelletization [16]. Pelletizing technology is mature from production to end-use and therefore pelletized CRDF can help with adopting it for wide-scale in waste management.”

The authors should add reference [xx] - Chemical Engineering Transactions, Volume 31, 2013, Pages 727-732 - both to the body text of the Introduction and the final list of references.

2) Materials and Methods - The authors should comment on the repeatability of their experiments.

3) Summary - The practical impact of the results obtained in this work should be better highlighted.

I’m willing to review the revised manuscript.

Author Response

(The authors gave the same response as above.)

Reviewer 3 Report

The manuscript reports on the mitigation of volatile organic compounds emissions from carbonized refuse derived fuel storage with pelletization. It provides important information in the area of solid fuel and gas emissions that is worthy investigation. However, English needs to be improved and some technical comments need to be addressed as follows:

1)Why the VOC emission was studied during a 4-day period? Why a longer period (2 weeks or 4 weeks) was not considered in this study? If this is done, should be added to the manuscript.

2)P1L37: This section should be modified from: 'Tests were carried out indicate that it is' to 'Tests indicated that'.

3)P1L40-41: In this statement (In this case, lower calorific value increased from 21.0 MJ.kg-1 [7] to 25.9 MJ.kg-1.), it is not clear why higher energy is obtainable? Because of torrefaction or because of pelletization? Pelletization does not increase the energy value per unit of mass!

4)P2L47: This needs to be revised: 'that >80 VOCs' to 'that more than 80 VOCs'

5)P2: Under Materials: Where this Original material (feedstock) for CRDF was obtained from? What was the properties in terms of chemical and physical properties? Please include in the text with details.

6)P2L73: A brief description of SPME method needs to be incorporated in this section for clarification to the reader.

7)P2L74: Technical information of the GC and MS instruments need to be included in this section.

8)What conditions of operation was used in GC? Please include in the section under Materials and Methods

9)What equipment was used for pelletization? What conditions? How pelletization was conducted? All need to be included in the text under Materials and Methods.

10)There is no discussion on the reason why pelletized CRDF generated much less emissions? This needs to be included in the text with reference.

11)In a figure, the most important compounds (4 or 5) from emission should be graphed and compared for both bulk and pelletized CRDF, for highlight of the results.

Author Response

We included detail responses to the reviewer in the attached file.

Round  2

Reviewer 1 Report

The authors improved greatly the manuscript with more references to other works and a better presentation of the overall work compared to the litterature. I took time to read and compare the new manuscript to the previous one, and appreciated the effort to present it more clearly. However, I still consider that it has to be rejected as an important amount of work is needed to improve the manuscript, as in my opinion it does not present sufficient novelty to be presented as is in a communication format.

Actually, my main concern is about the main result presented in the manuscript. Decrease of 86% of VOC emission after 4 days storage as already been published by the authors in a previous article (Bialowiec et al., Sustainibility, 2019, 11(3), 935). In the abstract of that article they say : "Pelletization significantly decreased (63%~86%) the maximum total VOC emission potential from stored CDRF.". The only difference between those publications concerns the list of VOC that was not published in the previous article. Proof of concept of emission reduction has then already been established, and the list and amount of VOC is not sufficient to be considered as novel work.

New experiments should be added to improve the interest and novelty of the manuscript. The value of 86% mitigation rate is possibly not reproducible as different type of RDF or storage conditions (humidity, O2 content,...) is bound to modify the VOC emissions. There is a need of more experiments to present mean values, and standard deviations. A blank has to be added to evaluate the amount of gas present in the air used in the glass vessel for the experiment (was it the same day? room air or standard N2/O2 pure air?).
Fire and explosion risks are mainly due to volatile such as H2 and CH4 that are not studied here, so conclusions of the experiment on that matter may be partial. Such mitigation rate values could greatly improve the interest to the reader.

Author Response

We provided the responses to the reviewer's comments in the attached file.

Reviewer 2 Report

The paper has been quite improved after revisions. However, there are two points that have not been adequately addressed.

1) The authors substantiated the first of the two sentences I suggested in my first comment (first round of review):

“Moreover, the storage, transport, and use of combustible dust pose severe safety issues, as such dust may explode when accidentally dispersed in air and ignited [18]. Thus, pelletization of CRDF is also useful to reduce the explosion risk [1].”

with reference [18] = Restuccia, F., Mašek, O.; Hadden, R.H.; Rein, G. Quantifying self-heating ignition of biochar as a function of feedstock and the pyrolysis reactor temperature. Fuel 2019, 236, 201-213, Doi: https://doi.org/10.1016/j.fuel.2018.08.141

and not with the reference I indicated (although they stated to have done so in the “Author Response” file).

The work by Restuccia et al. is a valuable contribution. However, this work is focused on self-heating ignition of bio-char, and not on explosion of combustible dusts that are accidentally dispersed and accidentally ignited by an external source. Thus, this work is fully inadequate to substantiate the sentence “Moreover, the storage, transport, and use of combustible dust pose severe safety issues, as such dust may explode when accidentally dispersed in air and ignited”.

Thus, I have still the following comments that the authors should scrupulously implement into the revised manuscript prior to publication:

A) They should write “dusts” and not “dust” in the sentence “Moreover, the storage, transport, and use of combustible dust pose severe safety issues, as such dust may explode when accidentally dispersed in air and ignited [18].”

B) They should substantiate the sentence “Moreover, the storage, transport, and use of combustible dusts pose severe safety issues, as such dusts may explode when accidentally dispersed in air and ignited [18].” with Ref. [18] = Chemical Engineering Transactions, Volume 31, 2013, Pages 727-732 (to be included in the final list of references).

2) Summary - The practical impact of the results obtained in this work should be better valorized.

I’m willing to review the revised manuscript.

Author Response

We provided the responses to the reviewer's comments in the attached file.

Round  3

Reviewer 1 Report

Dear authors,
The modifications made on the present manuscript helps to extend the reader interests for this work. I understand that there is a lot of work to be done to explore the path you are in, I guess that is why I felt frustated on my previous review and wanted to "know more".
The manuscript could then be published, after a few corrections :
l.42-43 : long spaces after kg-1
l.102 - "for preparation of the investment plans" => preparing ?
l.117-120: Could you clarify your idea ?
l.176- the following assumptions
l.192 workweek
l.205-208 : you identify four groups. Which one is the more proeminent ? Could you give the percentages of each group ?
l.209-230 : the list is not easy to read. There may be a better way to present it (in the table for example), and only give the number of compounds instead of a list.
l.236 : why are those 5 higher in pelletized than in ground CRDF ?
l.251-256 : Have you an idea of which VOC are impacted by this process ? Is there a trackers of that phenomenon ?
l.310 : requirements x2
l.316 : noted

Author Response

We submitted the responses to the reviewer's comments in the attached file.

Reviewer 2 Report

I have still the following comments that the authors must implement into the revised manuscript prior to publication.

1) Introduction (lines 61 and 62)

to replace the word “dust” with the word “dusts” in the sentence

“Moreover, the storage, transport, and use of combustible dust pose severe safety issues, as such dust may explode when accidentally dispersed in air and ignited [18].”

I have already suggested this change in the second round of review. In the response of the authors to this point, they wrote “Thank you for this comment. We corrected it.” However, this is not true: they did not implement my suggestion into the revised manuscript.

2) Introduction (lines 61 and 62)

to substantiate the sentence

“Moreover, the storage, transport, and use of combustible dusts pose severe safety issues, as such dusts may explode when accidentally dispersed in air and ignited [18].”

with [18] =  Chemical Engineering Transactions, Volume 31, 2013, Pages 727-732

instead of the current reference [18] = Restuccia et al., Fuel, 2019 (236) 201-213.

As I have already commented in the second round of review, the work by Restuccia et al. (Fuel, 2019 (236) 201-213) is a valuable contribution. However, this work is focused on self-heating ignition of bio-char, and not on explosion of combustible dusts that are accidentally dispersed in air and accidentally ignited by an external source. Thus, this work is fully inadequate to substantiate the sentence “Moreover, the storage, transport, and use of combustible dusts pose severe safety issues, as such dusts may explode when accidentally dispersed in air and ignited”.

In the response of the authors to this same comment, they wrote “Thank you for this comment. After the careful consideration of both works, we agree with this opinion. We replaced the original reference with Di Sarli et al. 2013 as fitting better in this context.” However, this is not true: they did not implement my comment into the revised manuscript.

3) References (lines 423-425)

to replace the current reference [18] = Restuccia et al., Fuel, 2019 (236) 201-213

with [18] = Chemical Engineering Transactions, Volume 31, 2013, Pages 727-732

4) Summary

to highlight the practical impact of their results in a more incisive manner.

I’m always willing to review the revised manuscript prior to publication.

Author Response

We submitted the responses to the reviewer's comments in the attached file.

Round  4

Reviewer 2 Report

The authors have addressed my comments in an adequate manner. Overall, the manuscript has been improved after revision. Therefore, it can be accepted for publication in Materials as it is.

Author Response

We would like to thank Reviewer for fruitful collaboration.